# Targeting the Gut Microbiota for Prevention and Management of Type 2 Diabetes

**DOI:** 10.3390/nu16223951

**Published:** 2024-11-19

**Authors:** Sabrina Donati Zeppa, Marco Gervasi, Alessia Bartolacci, Fabio Ferrini, Antonino Patti, Piero Sestili, Vilberto Stocchi, Deborah Agostini

**Affiliations:** 1Department of Biomolecular Sciences, University of Urbino Carlo Bo, 61029 Urbino, Italy; sabrina.zeppa@uniurb.it (S.D.Z.); a.bartolacci2@campus.uniurb.it (A.B.); piero.sestili@uniurb.it (P.S.); deborah.agostini@uniurb.it (D.A.); 2Department of Human Science for Promotion of Quality of Life, University San Raffaele, 00166 Rome, Italy; vilberto.stocchi@uniroma5.it; 3Sport and Exercise Sciences Research Unit, Department of Psychology, Educational Science and Human Movement, University of Palermo, 90144 Palermo, Italy; antonino.patti01@unipa.it

**Keywords:** gut microbiota, type 2 diabetes, glucagon-like peptide-1, diet, supplements, lifestyle intervention, physical exercise

## Abstract

Type 2 diabetes (T2D) is a chronic metabolic disorder with a heterogeneous etiology encompassing societal and behavioral risk factors in addition to genetic and environmental susceptibility. The cardiovascular consequences of diabetes account for more than two-thirds of mortality among people with T2D. Not only does T2D shorten life expectancy, but it also lowers quality of life and is associated with extremely high health expenditures since diabetic complications raise both direct and indirect healthcare costs. An increasing body of research indicates a connection between T2D and gut microbial traits, as numerous alterations in the intestinal microorganisms have been noted in pre-diabetic and diabetic individuals. These include pro-inflammatory bacterial patterns, increased intestinal permeability, endotoxemia, and hyperglycemia-favoring conditions, such as the alteration of glucagon-like peptide-1 (GLP-1) secretion. Restoring microbial homeostasis can be very beneficial for preventing and co-treating T2D and improving antidiabetic therapy outcomes. This review summarizes the characteristics of a “diabetic” microbiota and the metabolites produced by microbial species that can worsen or ameliorate T2D risk and progression, suggesting gut microbiota-targeted strategies to restore eubiosis and regulate blood glucose. Nutritional supplementation, diet, and physical exercise are known to play important roles in T2D, and here their effects on the gut microbiota are discussed, suggesting non-pharmacological approaches that can greatly help in diabetes management and highlighting the importance of tailoring treatments to individual needs.

## 1. Introduction

The gut microbiota is a complex ecosystem within the gastrointestinal tract environment, comprising bacteria, viruses, fungi, archaea, and protozoa [1]. It was previously believed that there were almost ten times as many bacterial cells as human cells [2]; however, current research indicates that there are actually about the same number of them [3]. Gut bacteria are primarily composed of five phyla: 90% of them are represented by *Firmicutes* and *Bacteroidetes*, the remaining 10% mainly by *Actinobacteria*, *Proteobacteria*, and *Verrucomicrobia* [4]. Great interindividual differences in gut microbial composition exist, and this makes each person’s bacterial community distinct and unique [5]. This complex and dynamic ecosystem is modulated by not modifiable factors, such as host genetics and sex, and modifiable ones such as environmental factors, medications, and lifestyle choices, which affect diversity and richness [6,7]. There is mounting evidence that the gut microbiota is essential to human health and that an “imbalance” in the gut microbial community, namely dysbiosis, is linked to a number of clinical conditions, including autoimmune disease [8], inflammatory intestinal diseases [9], mental disorders [10], and metabolic diseases [11]. Obesity, non-alcoholic fatty liver, insulin resistance (IR), and chronic inflammation, which are influenced by the gut microbiota, are associated with the development of diabetes [12,13].

Diabetes mellitus (DM) refers to a range of metabolic dysfunctions marked by chronic hyperglycemia due to impaired insulin efficacy, insufficient insulin secretion, or both of them [14]. Diabetes can be classified into general categories, which are gestational diabetes, type 1 (T1D) and type 2 diabetes (T2D), and specific types of diabetes [15]. Gestational diabetes is a glucose use disorder first diagnosed during pregnancy [14,15]. In T1D, antibodies are generated against several components of pancreatic β-cells, which results in the islets that make insulin deteriorating and ultimately being totally destroyed, leading to a shortage of insulin [16]. This type also includes checkpoint inhibitor-induced diabetes and latent autoimmune diabetes in adults (LADA) [14,15]. T2D is characterized by non-autoimmune insulin deficiency associated with IR and frequently metabolic syndrome, while specific types of diabetes are due to several causes, such as genetic defects, genetic syndromes, disease of the exocrine pancreas, endocrinopathies, infections, and drug use [15]. The International Diabetes Federation (IDF) reported that, worldwide, 536.6 million adults (20–79 years), which represented 9.2% of the total, were diabetic in 2021, and T1D affected an additional 1.2 million children and adolescents aged 0 to 19 [17]. Furthermore, by 2045, the number of diabetics is expected to reach 700 million [18]. More than 90% of cases of DM are T2D, a disorder characterized by non-autoimmune insulin deficiency associated with IR and often metabolic syndrome, often undiagnosed for many years but with considerable associated risks [15].

For a large population, this is a serious issue because T2D can cause severe comorbidities, including microvascular and macrovascular complications, accounting for the majority of patient morbidity and death [19,20]. It is important to note that prediabetes, a condition characterized by slightly elevated blood glucose level, is also associated with cardiovascular events and death [21], confirming that these complications can arise before T2D develops. T2D complications cause patients’ physical and emotional suffering in addition to a significant financial burden on medical care. This chronic disease’s development is heavily influenced by both hereditary and environmental factors interacting with each other and determining T2D prevalence. Environmental factors that contribute to its ongoing rise in prevalence include diet, ambient air pollution, and physical inactivity [22,23]. In terms of the pathophysiology of T2D, IR causes the liver to produce more glucose and absorb less from the muscles. Additionally, IR-induced β-cell dysfunction lowers insulin secretion, creating a feedback loop that leads to hyperglycemia and further exacerbates the condition. As obesity is linked to metabolic abnormalities that cause IR, it is a significant risk factor for T2D. Engaging in physical exercise helps to prevent obesity and acts on insulin receptors to increase muscle glucose uptake from plasma [24].

Some of the suggested causes for impaired insulin production and sensitivity in T2D include oxidative stress, endoplasmic reticulum stress, ectopic lipid deposition, amyloid deposition in the pancreas, lipotoxicity, and glucotoxicity. Oxidative stress, defined as an imbalance between the body’s antioxidant defense systems and the generation of reactive oxygen species (ROS), plays a key role in the etiology, development, and consequences of T2D. Hyperglycemia and mitochondrial dysfunction associated with T2D promote ROS production, and oxidative stress is known to enhance IR and decrease β-cell activity, thereby altering glucose homeostasis. Oxidative stress can harm any organ or tissue in the body, leading to diabetic retinopathy, neuropathy, and nephropathy and resulting in vascular consequences and cardiovascular illnesses [25]. ROS are produced more readily when oxidative stress triggers the release of inflammatory mediators. Numerous studies have demonstrated that IR and the traits of metabolic syndrome, such as hyperglycemia, are connected to sub-clinical inflammation and that low-grade inflammation increases the risk of T2D, which can be considered an inflammatory disease [26]. In order to understand the complex pathophysiology of this multifactorial disease, the alteration of metabolic interorgan crosstalk (the signaling between tissue and secreted factors) should also be considered. Hormones, insulin, glucagon, glucagon-like peptide-1 (GLP-1), organokines, and other metabolic molecules, including amino acids, lipids, and free fatty acids (FFAs), are among the regulators known to influence metabolism that are involved in T2D development and progression [27].

The large number of individuals who are at risk for T2D, or suffer from it, suggests the need to look for further reasons for the pathophysiology of the illness. This can result in the development of strategies aimed at reducing its risk and novel treatments targeted at stopping or reversing the disease’s progression. The aim of this review is to explore the complex relationship between the gut microbiota and T2D, highlighting how an integrated and individualized approach, based on diet, exercise, and supplementation, could help in disease prevention and management.

## 2. Interaction Between the Gut Microbiota and T2D

The gut microbiota has been demonstrated to be altered in diabetic patients, indicating its significant role in the pathophysiology of T2D [28,29,30]. A fecal metagenomic analysis in European women revealed that T2D patients had four *Lactobacillus* species that were enriched and five additional *Clostridium* species that were decreased in abundance compared to healthy individuals [31]. It should be mentioned that the *Lactobacillus* species, which could be associated with augmented glucose presence in the intestine, correlated positively with hemoglobin A1c (HbA1c) and fasting glucose, whereas the *Clostridium* species linked negatively with insulin, plasma triglycerides, fasting glucose, and HbA1c and positively with adiponectin and high-density lipoprotein (HDL). Furthermore, prototypical butyrate producers *Faecalibacterium prausnitzii* and *Roseburia intestinalis* were very discriminant for T2D [31]. The authors also reported that metagenomic markers for T2D differed between people of different ages and population, suggesting that they greatly depend on the context and should be better considered as a whole [31]. It has also been discovered that metabolic parameters are linked to the changed gut microbiota in T2D. For example, a strong and positive correlation exists between the ratio of *Bacteroidetes* to *Firmicutes,* the ratio of *Bacteroides-Prevotella* group to *C. coccoides-E. rectale* group, and plasma glucose levels [32]. *Fusobacterium*, *Ruminococcus*, and *Blautia* were positively correlated with T2D, whereas *Bacteroidetes*, *Bifidobacterium*, *Akkermansia*, *Faecalibacterium*, and *Roseburia* were negatively correlated [33]. It should be underlined that *Akkermansia* are a very beneficial bacteria for host metabolism, known for protecting the gut barrier. Interestingly, prediabetics also have not-common gut microbiota [34,35,36]; furthermore, the microbial composition varies across phases of T2D.

Although the role of individual bacterial species in promoting or counteracting T2D has not yet been clarified, except in a few cases, the reported studies provide a general picture of the gut microbiota in diabetics and healthy people. It should always be kept in mind that the microbiota is an ecosystem that affects its host and can be greatly influenced by lifestyle. It has been shown that, in obesity and related metabolic illnesses, dietary modifications quickly cause alterations in both *Firmicutes* and *Bacteroidetes* levels. Hwang et al. demonstrated that depletion of *Firmicutes* and *Bacteroidetes* caused by antibiotics in diet-induced obesity in mice alleviated systemic glucose intolerance, hyperinsulinemia, and IR via GLP-1 production [37]. In response to a meal, enteroendocrine L cells release GLP-1, an insulinotropic hormone that is essential for blood sugar levels, controlling secretions of insulin and glucagon, emptying of the stomach, blood flow, and food consumption. GLP-1 is quickly degraded and deactivated by the enzyme dipeptidyl peptidase 4 (DPP4) [38] and acts in a paracrine manner by binding locally to its receptor in the vagus nerve and enteric nervous system, stimulating the gut–brain axis, and in an endocrine manner by triggering β-cells [39]. Currently, GLP-1 receptor agonists like liraglutide, lixisenatide, exenatide, dulaglutide, albiglutide, and semaglutide are used for the therapy of T2D [40]. Through metabolites that control enteroendocrine cells (EECs) and, subsequently, hormonal activity, the gut microbiota influences host GLP-1 synthesis [41]. Furthermore, GLP-1 release was more prompt in the morning than in the afternoon in humans, indicating that GLP-1 secretion possesses temporal variability [42]. The gut microbiota appears to be crucial in regulating the rhythm of GLP-1 secretion since insulin production did not follow a diurnal pattern in germ-free animals lacking a gut microbiome, but it does after fecal transplantation from mice consuming a regular diet [43]. GLP-1 is stimulated by metabolites from the gut microbiota, but the gut microbiota is also regulated by GLP-1 through appetite, satiety, nutrient regulation, and mechanisms involving inflammatory responses [41]. Below, several mechanisms that elucidate the close relationship between T2D and gut bacteria are described, and the strategies that can help in obtaining a healthier microbial profile are reported.

### 2.1. Gut Barrier Permeability and Inflammation

The intestinal epithelium barrier interacts strongly with immune system cells and the gut microbiota rather than acting as a static physical barrier. The gut microbiome’s composition influences immune system development and modifies immunological mediators, both of which have an impact [44] on the intestinal epithelium. Dysbiosis, which is an imbalance of gut microbial species, may also promote intestinal barrier breakdown and be linked to a higher risk of contracting specific illnesses [45]. The leaky gut syndrome (LGS) hypothesis states that intestinal hyperpermeability could permit pathogens and toxins to enter through intestinal epithelium junctions into the bloodstream, impacting immune, respiratory, reproductive, hormonal, or neurological systems [45]. The intestinal protective layer serves as a barrier between the body and gut contents; when it malfunctions, bacteria or bacterial products can seep into the body, increasing the risk of T2D [46]. The lipopolysaccharide (LPS) produced by gut bacteria causes metabolic endotoxemia and IR in addition to acting as a trigger for chronic low-grade inflammation through the LPS-CD14 pathway [47]. Furthermore, studies conducted in vitro and in vivo confirmed that hyperglycemia alters the integrity of tight and adherent junctions, increasing intestinal barrier permeability and allowing microbial compounds to enter into the bloodstream [48]. A key role for GLP-1 in suppressing inflammation and restoring mucosal integrity has been suggested by Lebrun et al.’s demonstration that, after gut barrier injury, enteroendocrine L cells recognize LPS, enhancing the release of GLP-1 via a Toll-like receptor 4 (TLR4)-dependent mechanism, which precedes the onset of significant modifications in inflammatory status and LPS levels [44]. Enhancing gut barrier function and ameliorating inflammation can greatly help in counteracting T2D and its complications. Several studies confirmed improved gut permeability and reduced LPS infiltration using *Akkermansia muciniphila*, a bacterial strain usually with reduced abundance in diabetes. *A. muciniphila* has been reported to enhance *Mucin 2* gene expression, strengthening mucus barrier function, and to preserve gut cell layer health, promoting intestinal epithelium renewing and tight junction protein expression. Furthermore, its abundance positively correlates with anti-inflammatory cytokines and negatively with pro-inflammatory factors [49]. Given its protective effect against endotoxemia, combined with its ability to improve metabolism and maintain microbiota homeostasis, *A. muciniphila* has recently been proposed as a “next-generation probiotic” for alleviating metabolic disorders [49].

### 2.2. Short-Chain Fatty Acids (SCFAs)

The intestinal microbiota breaks down and ferments dietary fibers that are ingested with food, which aids in the metabolism of carbohydrates, producing monosaccharides and short-chain fatty acids (SCFAs), which become important energy sources for the host and can act as substrates for lipogenesis and gluconeogenesis. The principal SCFAs are acetic acid, propionic acid, and butyric acid. According to earlier research, diabetic individuals have less bacteria that produce SCFAs, and consequently less SCFAs produced [50]. Dietary fibers are mostly broken down by *Bacteroides*, *Prevotella*, *Parabacteroides*, and *Alistipes* species belonging to the phylum of *Bacteroidetes.* In addition to having a local effect on the colon and acting as energy source, SCFAs have the ability to enter the bloodstream and alter the metabolism of other organs. Their mode of action involves the stimulation of G protein-coupled receptors (GPCRs), such as GPR43 and GPR41, which are expressed in a number of different tissues, including adipose tissue, distal ileum, colon, lymph nodes, and the inhibition of histone deacetylases (HDAs), causing gene transcription and metabolic modifications with a pleiotropic effect [51,52]. Specifically, SCFAs can act in T2D prevention and management in several manners, for example, preserving the integrity of the gut epithelium by stimulating the synthesis of mucin [53] and enhancing the host immune system by affecting the activities of macrophages [54], preventing the entry of pathogens. Furthermore, acetate can promote immunometabolism, acting as an energy source for immune cells [55]. Interestingly, SCFAs, through GPR43 and HDAs, promote Peptide-YY (PYY) and GLP-1 expression and/or secretion, regulating food intake and preventing obesity [56,57]. Additionally, it was demonstrated that human adipose-derived mesenchymal stem cells expressed GPR43 and that upon propionate binding, these cells’ reduction in lipid accumulation indicated the prevention of adipogenesis [58]. This obesity prevention effect represents another antidiabetic action of SCFAs.

Diabetics exhibit lower numbers of SCFA-producing bacteria and circulating SCFAs compared to healthy individuals, with fewer genes involved in SCFA synthesis due to microbiota disturbance. Specifically, reduced levels of butyrate and butyrate-producing bacteria, such as *Faecalibacterium prausnitzii*, are associated with T2D due to increased inflammation. This bacterium contributes to butyrate production and has shown potential in reducing IR by inducing GLP-1 secretion through the fatty acid receptor FFAR2 [59]. The butyrate biosynthetic pathway of *F. prausnutzi* is shared by *Roseburia* spp., hypothesized to be a marker of health due to the decrease observed in several diseases, including T2D [60]. Butyrate has been shown to slow down the onset of T2D via a variety of processes, including preserving the integrity of the intestinal epithelial barrier [61], enhancing insulin sensitivity, and reducing inflammation and appetite. Butyrate is also important for transcriptional regulation and post-translational modifications, as it significantly inhibits lysine and HDA activity [62]. This inhibition results in histone hyperacetylation, increasing the accessibility of transcription factors to gene promoter regions. Additionally, butyrate acts as a ligand for two transcription factors: peroxisome proliferator-activated receptor γ (PPARγ) and the aryl hydrocarbon receptor [63,64]. For these reasons, several strategies have been described to increase butyrate levels in T2D patients who are often butyrate-producing bacteria deficient [65].

### 2.3. Branched-Chain Amino Acids (BCAAs)

In addition to SCFAs, functional products of the gut microbiota, such as amino acids, are linked to the pathophysiology of obesity, IR, and T2D. The gut microbiota can affect the amount of amino acids that are bioavailable from either endogenous or alimentary proteins and can thus provide the host with amino acids [66]. Crucially, new research indicates that increased systemic concentrations of branched-chain amino acids (BCAAs), namely leucine, isoleucine, and valine, in adult humans are associated with obesity, IR, and T2D [67] and might be thought as markers of IR and predictors of the onset of T2D. Zou et al. offered proof-of-concept data for the therapeutic feasibility of modifying BCAA metabolism to treat diabetes and showed that a BCAA catabolic deficiency and increased abundance of BCAAs and BCKAs play a crucial causative role in obesity-associated IR [64]. Moreover, Liu et al. revealed that branched-chain α-keto acids (BCKAs), which are endogenous metabolites of BCAAs, might exacerbate inflammation and organ damage in T2D by inducing mitochondrial oxidative stress and cytokine release in macrophages [68]. A positive correlation was found between the amounts of BCAAs, SCFAs, *Prevotella*, *Alistipes*, and *Barnesiella*; however, a negative correlation was observed with *Bacteroides* and *Enterococcus*. It is interesting to note that *Prevotellaceae* have been discovered to be considerably enriched in obesity [69]. Yoshida et al. demonstrated a favorable correlation between the gut microbiota and BCAA catabolism in brown adipose tissue, which is a modulator of metabolic and cardiovascular disorders [70]. Through an anti-inflammatory action, they demonstrated that treatment with *Bacteroides* spp. like *B. dorei* and *B. vulgatus* reduced obesity-induced BCAA catabolic deficiencies in this tissue and, therefore, obesity, underscoring the significance of specific bacterial species in BCAA metabolism and metabolic health [70].

### 2.4. Bile Acids (BAs)

The liver produces primary bile acids (BAs) when cholesterol is broken down, which are involved in the uptake and transportation of fats and fat-soluble vitamins. Cell death, apoptosis, and inflammation can all be brought on by toxic bile acids. Conversely, bile acid-stimulated nuclear and GPCR signaling protect against hepatic, intestinal, and macrophage inflammatory responses. Depending on where receptor activation occurs—in the intestine or the liver—it may either prevent or contribute to steatosis and obesity [71]. Abnormal concentrations of BAs and fecal metabolites have been reported in people suffering of metabolic disorders and in T2D patients [72]. In this context, the gut microbiota plays a central role since it converts primary BAs into secondary BAs when they reach the colon [73], preventing accumulation. An impairment in the conversion from primary to secondary BAs may contribute to intestinal inflammation since secondary BAs can exert anti-inflammatory effects [74]. In addition, certain secondary BAs preserve the intestinal tract’s barrier function by impeding the development and migration of gut bacteria into host cells [75]. Deoxycholic acid and lithocholic acid can also be harmful to the host, causing oxidative stress, membrane damage, and colonic carcinogenesis [76], emphasizing the various and different functions of microbial-generated secondary BAs. Finally, secondary BAs can either activate the intestinal L cell’s Takeda G protein-coupled receptor 5 to increase secretion of GPL-1 [77] or inhibit it by the farnesoid X receptor (FXR) [78], meaning they have dual regulatory effects on GPL-1 secretion. On the other hand, BAs are strong antibacterial substances that are differently tolerated by bacteria and have a significant impact on the gut’s microbial ecology [79]. For this reason, the complex metabolism of BAs and the dual role of secondary BAs in gut health should be considered in evaluating the gut microbiota’s effects on T2D.

### 2.5. Tryptophan and Its Metabolites

Tryptophan is an essential aromatic amino acid that is obtained from foods, including milk, cheese, chicken, fish, and oats. Tryptophan plays a key function in the production of proteins and is a precursor to several important metabolites. Even if most of the tryptophan in food is processed locally in the gut by host enzymes, gut microorganisms digest around 5% of it. The three main metabolic routes for tryptophan are kynurenine (KYN) and its derivatives, indole, and serotonin [80].

IR and T2D are linked to tryptophan dysmetabolism. KYN and KYN metabolite production is correlated with the inflammatory status in metabolic syndrome through the activation of indoleamine 2,3-dioxygenase 1. Concurrently, there is a deficiency in the gut microbiota’s ability to produce indole-3-propionic acid (IPA) and other ligands for the aryl hydrocarbon receptor (AhR). Inadequate activation of the AhR pathway modifies gut permeability and facilitates LPS translocation by reducing GLP-1 and interleukin (IL)-22. Furthermore, gut bacteria constitute a crucial modulator of the biosynthesis of serotonin, which affects food behavior and satiety, and in consequence, body mass index (BMI), and is disrupted in metabolic syndrome [81]. Chimerel et al. demonstrated that indole is important for gut microbiota–mouse colonic L cells communication. Indole, on the one hand, it caused an acute stimulation of GLP-1 secretion by inhibiting voltage-gated K^+^ channels and improving Ca^2+^ entry during short exposure; on the other hand, indole decreased the rate at which ATP was created by inhibiting NADH dehydrogenase, which led to a long-lasting reduction in GLP-1 secretion over longer period [82]. In this context, diet plays an important role. For instance, increased intake of fiber and milk (which have higher levels of gut *Bifidobacterium*) is linked to a better-circulating tryptophan metabolite profile with enhanced IPA synthesis [83]. These data underline the importance of a targeted diet in achieving an equilibrium microbial state.

### 2.6. Trimethylamine N-Oxide (TMAO)

L-carnitine and phosphatidylcholine-rich foods are broken down by gut bacteria into trimethylamine, which the liver then converts to trimethylamine N-oxide (TMAO). Greater TMAO concentrations were related to T2D and increased mortality risk independent of glycemic control in T2D patients [84]. Gao et al. demonstrated that, in mice given a high-fat diet, TMAO worsened poor glucose tolerance, blocked the hepatic insulin signaling pathway, and induced inflammation in adipose tissue [85]. It is worth noting that individuals with dysbiosis produce more TMAO than those with eubiosis while ingesting the same food [86]. Increasing data suggest that TMAO is linked to increased risks of cardiovascular disease and renal failure. In mice, dietary treatment with TMAO, carnitine, or choline modified the cecal microbial composition, raising the risk of atherosclerosis [87]. Strategies to reestablish eubiosis and certain dietary adjustments might help not only in counteracting the onset and progression of T2D but also the comorbidities, such as cardiovascular diseases, related to it.

### 2.7. Antidiabetic Drugs

T2D and obesity are known to induce gut dysbiosis, while antidiabetic drugs can help to restore a healthy microbiota [88]. Alpha-glucosidase inhibitors slow carbohydrate uptake in the distal intestine, favoring bacterial growth. The most commonly used of these inhibitors, Acarbose, increases the number of taxa that produce SCFAs, such as *Lactobacillus*, *Faecalibacterium*, *Prevotella* [89], and *Bifidobacterium* [90]. Sitagliptin, which is a dipeptidyl peptidase IV inhibitor, as well as metformin, commonly used to lower glucose and enhance insulin sensitivity, have been demonstrated to increase the relative abundance of the genus *Lactobacillus* [90]. In T2D complicated with nonalcoholic fatty liver disease, the intestinal bacterial community exhibited significant increases in diversity and richness upon receiving treatment with liraglutide, a GLP-1 receptor agonist. Additionally, the relative abundances of *Bacteroidetes*, *Proteobacteria*, and *Bacilli* were significantly elevated, while those of *Fusobacteria* and *Actinobacteria* were significantly enhanced upon metformin treatment [91]. Thus, different antiglycemic drugs exert different effects on the composition of the gut microbiota, selectively increasing beneficial bacteria [90]. It is known that the efficacy of drugs greatly depends on gut microbiota features, and gut microbiota modulation could influence therapy outcomes [92]; for example, liraglutide responsiveness is likely to depend on gut microbiota dysbiosis [93]. Restoring eubiosis in the altered gut microbiota in T2D patients can greatly help in potentiating the effect and reducing the doses of traditional drugs.

## 3. Diet, T2D, and the Gut Microbiota

An unbalanced and unhealthy diet is one of the primary risk factors for chronic illness since it is widely acknowledged that eating habits have a substantial influence on our overall health and wellbeing. Diet is one of the most important lifestyle interventions in T2D prevention and management. A number of dietary strategies, such as a low-carbohydrate diet (LCD), very-low-calorie diet (VLCD), fasting-mimicking diet (FMD), and Mediterranean diet (MedDiet), have been suggested to reduce the occurrence of chronic diseases [94].

### 3.1. Low-Carb, Very-Low-Calorie, and Fasting-Mimicking Diets

Given their positive effects on weight reduction and glycemic management, research shows that low-carb (<130 g/day of carbs) and very low-carb diets (usually <50 g/day of carbs) can be useful strategies for treating T2D. In clinical trials and standard care, LCDs have shown weight and HbA1c-lowering effects [95]. A primary care practice in England using an LCD approach saw 46% of people with T2D achieve remission, with improvements in lipid profile, blood pressure, and weight loss. International researchers defined T2D remission as HbA1c <6.5% at 3 months post antidiabetic medication cessation, without bariatric surgery [96]. Significant weight loss through LCDs and VLCDs led to a 25–77% remission rate in overweight and obese individuals, with reduced HbA1c and decreased medical support [97,98]. Higher remission rates were directly associated with greater weight reduction, according to recent results from the Diabetes Remission Clinical Trial (DiRECT). Individuals who lost more than 10 kg were far more likely to achieve remission than those who lost less. People with T2D experienced a remission rate of 46–60% after one year [99]. Significant weight reduction was accomplished using an LCD. Consequently, 20% of all practicing T2D patients achieved remission. It seems that a time frame of less than one year for T2D treatment constitutes a significant window of opportunity for reaching drug-free diabetic remission [100,101].

Studies show that most patients experienced remission because of dietary strategies, with significant changes in blood lipids, HbA1c, blood glucose, and quality of life. Remission was more likely in those who lost more weight and had a shorter T2D diagnosis duration [102]. VLCD caused weight loss by 20–30%, sometimes in just 12–16 weeks when daily energy intake was restricted to 400–800 kcal/day [103]. Steven et al. reported that for the 40% of patients who reacted to a VLCD by reaching a fasting plasma glucose of <7 mmol/L, a comprehensive and long-lasting weight reduction program resulted in a continuous remission of T2D for at least 6 months [104]. Currently, available data indicates that only people who have been diagnosed with T2D relatively recently may benefit from VLCD or other calorie restriction techniques to induce T2D remission.

While there is evidence that extensive behavioral support can increase the durability of weight reduction with VLCD, people who do not receive the best possible assistance for sustained behavior change will still find it difficult to successfully adhere to these dietary methods over the long term [103]. In T2D patients using diet alone or metformin for glycemic management, Van der Butg et al. found that a 5-day monthly fasting-mimicking diet over a 12-month period increased HbA1c levels while requiring fewer pharmaceuticals, looked safe in standard clinical practice, and decreased the need for glucose-lowering drugs [105].

### 3.2. Mediterranean Diet

Among diets, MedDiet, one of the world’s most popular and researched regimens, has been linked to several health advantages [106]. The Mediterranean diet provides several benefits on metabolic illness and T2D risk. Molecules within the Mediterranean diet (MedDiet) can counteract mechanisms involved in T2D progression, including inflammation and oxidation. In particular, MedDiet, adding another piece to the puzzle of metabolic pathology prevention, can modulate the gut microbiota. A diet that emphasizes eating more fruits, vegetables, whole grains, and seafood while consuming fewer starches, sugary beverages, and red and processed meats can postpone the onset of T2D, according to epidemiological research [107]. Moreover, a high intake of dietary fiber (particularly cereal fiber), antioxidants, and monounsaturated fatty acids (MUFAs), along with foods containing these nutrients, has been linked to improved insulin sensitivity, the ability of pancreatic β-cells to secrete insulin, and a lower risk of developing T2D (Figure 1), according to consistent epidemiological and clinical evidence [108,109]. MedDiet is praised for its antidiabetic properties, making it a suitable alternative to low-fat, high-carb diets for managing blood glucose in T2D patients [110].

By increasing antioxidant capacity and decreasing inflammation, Filippatos et al. discovered that moderate to high adherence to MedDiet could prevent T2D [111]. More adherence to MedDiet might reduce the incidence of T2D by 19%, according to a meta-analysis of trials including 122,810 individuals [112]. High-MUFA diets have been demonstrated in recent years to have a beneficial impact on glycemic management and total triglyceride (TG) levels in T2D patients [109]. Additionally, two prospective studies have connected the risks or outcomes of gestational T2D to MedDiet adherence. Participants in the highest quartile of MedDiet adherence had a 40% reduced chance of developing T2D compared to those in the lowest quartile, according to a nurses’ health survey that found 491 occurrences of incident T2D among 4413 females followed for 14 years and aged 22–44 with a history of gestational diabetes [113]. Adherence to the Mediterranean diet (MedDiet) was associated with a lower prevalence of gestational diabetes and better glucose tolerance, even in women without the disease, according to research including 1076 pregnant women from ten different countries [113]. Thus far, the vast majority of research has evidenced the value of MedDiet in T2D prevention. What is even more exciting is the fact that studies on pregnant women have increased the group for whom MedDiet adoption is appropriate for T2D prevention. MedDiet education, depending on the population size, may be a safe public health strategy to stop or postpone the onset of T2D. To elucidate the mechanisms of T2D risk reduction that are not reliant on weight loss, more study is needed [106], but it is likely that most of the diet-induced benefits in managing T2D pass through gut microbiota modulation. Mediterranean diet components have an important impact on the gut microbiota’s development. The major intake of fiber, vitamins, antioxidants, and mono- and poly-unsaturated fatty acids and minor consumption of processed foods, saturated fatty acids, and amino acids induce an increase in microbiome diversity.

### 3.3. Effects of Diet on the Gut Microbiota

According to human studies, MedDiet can change the composition of gut microbes. Donati Zeppa et al. reported a significant decrease in the abundance of *Proteobacteria* in 20 BC survivors of the MoviS clinical trial following a 12-week home-based lifestyle intervention based on MedDiet and exercise [114]. The adherence of eating habits to MedDiet led to a reduction of the abundances of *Proteobacteria* and *Firmicutes* and an increase in the abundance of advantageous bacterial communities such as *Bacteroidetes*, *Lactobacilli*, *Bifidobacteria*, and *Faecalibacterium*. These bacterial modifications associated with enhanced production of microbiota-derived metabolites have been observed in human studies. SCFAs and MedDiet itself lead to oxidative stress, inflammation, obesity, and TD2 reductions [115].

Wang et al. [116] reported the potential connections between the gut microbiome, MedDiet, and T2D. Participants in this study comprised 394 normoglycemic, 805 prediabetic, and 543 diabetic individuals from a cohort study of Hispanic/Latino men and women in the United States. Increased adherence to MedDiet was linked to increased abundance of major dietary fiber metabolizers, such as *Faecalibacterium prausnitzii*, *Coprococcus*, and *Lachnospira*. The majority of adherents to MedDiet were found to have more or less abundant microbial activities linked to the metabolism of amino acids and carbohydrates, with the exception of those involved in the breakdown of galactose and lactose. A depletion of bacteria related to dietary sulfur reduction to H_2_S has also been associated with the higher adherence to MedDiet. In particular, MedDiet showed a stronger protective effect against T2D in participants with lower levels of *Prevotella*. The microbial adaptations induced by MedDiet led to increased microbiota-mediated metabolites, intestinal homeostasis, decreased dysbiosis, and decreased intestinal permeability, contributing to the prevention of T2D onset [117].

Previous analysis has indicated a correlation between eating vegetarian and improvements in T2D glycemic management [118]. In this context, Panhigrai et al. [119] proposed a lifestyle intervention that promoted adherence to a plant-predominant diet in T2D patients. This diet was rich in phytochemicals and bioactive nutritional molecules, such as vitamins and fiber, and low in fat. A dietary fiber intake of 50 g per day (25 g soluble and 25 g insoluble fiber) induced an improvement in glucose homeostasis and insulin secretion in T2D patients, probably due to the increase in SCFA products by the anaerobic microbial fermentation of fiber. Total SCFA, acetate, and butyrate levels in human feces were found to be negatively associated with the risk of developing T2D in cross-sectional research from the Henan rural cohort [120]. SCFAs stimulate GLP-1 and PYY in the stomach, which improves glucose homeostasis [121]. The authors also identified SCFA interval levels associated with decrease in T2D prevalence, although several studies will be needed to identify a healthy individual SCFA concentration to prevent T2D. Whole grains and legumes are known to lower postprandial blood glucose levels, both during the meal they are consumed and at subsequent meals. This has significant consequences for blood glucose regulation throughout the day and the avoidance of T2D [122].

In conclusion, more accurate and successful dietary interventions for the prevention of T2D should arise from the personalization of diets based on the gut microbial compositions of each individual.

## 4. Supplements, T2D, and the Gut Microbiota

Research on animals suggests that the evolution of IR to T2D may be influenced by the nature of intestinal microbes. By lowering intestinal endotoxin concentrations, altering the structure of the gut microbial population, and lowering energy harvest, probiotics and/or prebiotics may be a viable strategy for improving insulin sensitivity [123]. Probiotics have antidiabetic properties via decreasing oxidative stress, intestinal permeability, and pro-inflammatory cytokines through the nuclear factor kappa-light-chain-enhancer of activated B cells (NF-κB) pathway. To create guidelines for the use of pro- and prebiotics in the prevention of T2D, better-planned human clinical trials are needed (Table 1). Furthermore, other nutraceuticals, such as antioxidants and traditional Chinese medicine, have been reported to exert antidiabetic activity, although most of them need to be metabolized and activated by gut microorganisms.

### 4.1. Probiotics

Probiotics are live microorganisms that, when administered in the appropriate amounts and for sufficient duration, can benefit the host’s health [124]. They work by reducing inflammation, strengthening the intestinal barrier, and ultimately restoring gut health. Probiotics are currently undergoing extensive research as potential biotherapeutics thanks to their health-promoting properties and their capacity to combat certain diseases [125]. Given the gut microbiota characteristics in patients with T2D and the resulting consequences, probiotics are recognized as beneficial and complementary. Probiotics usually improve T2D symptoms, enhancing intestinal integrity and restoring the host’s intestinal barrier function through their surface molecules and metabolites, reducing systemic LPS levels, lowering endoplasmic reticulum stress, and improving peripheral insulin sensitivity [126]. Probiotics, specifically *Lactobacilli* and *Bifidobacteria*, have recently gained attention as promising biotherapeutics with proven efficacy demonstrated in different experimental models. Some studies have reported reductions in the abundances of *Bifidobacterium* spp. and *Lactobacillus* spp. with increased plasma LPS, which caused metabolic endotoxemia via NF-kB activation at the molecular onset of IR. One hypothesis is that T2D can be improved by decreasing the concentration of LPS in the blood [125,127]. Ma et al. investigated the activation of NF-kB signaling in the progression of inflammation using a HeLa cell line preincubated with live *Lactobacillus reuteri* cells for 1–2 h. This treatment prevented translocation of NF-kB to the nucleus and inhibited the expression of various pro-inflammatory cytokines regulated by NF-kB [128].

In a double-blind, randomized, placebo-controlled trial, 50 volunteers consumed 120 g/d of fermented milk daily for 6 wks. The subjects were divided into two groups: the probiotic group, who consumed fermented milk containing 109 colony-forming units/d of *Bifidobacterium animalis* subsp lactis BB-12 and *Lactobacillus acidophilus* La-5, and the control group, who had regular fermented milk. After 6 weeks, the intervention arm showed an improvement in their glycemic profile. There was a significant decrease in fructosamine levels, and HbA1c levels also appeared to be reduced. By contrast, the control group did not show significantly affected glycemic control [129]. In addition, *A. muciniphila* has a beneficial role in glycemic control and IR. Furthermore, its important roles in promoting intestinal barrier function, lowering inflammation, boosting metabolism, and preserving microbial homeostasis have received recent research interest [49]. Everard et al. proved that treatment with *A. muciniphila* for 4 weeks reversed metabolic disorders induced by a high-fat diet. This included fat mass gain, metabolic endotoxemia, adipose tissue inflammation, and IR in male C57BL/6 *ob*/*ob*, HF-fed obese, and T2D mice. *A. muciniphila* administration increased the levels of endocannabinoids in the intestine, which control inflammation, gut barrier function, and gut peptide secretion [130].

Several studies demonstrated the effect of probiotics in GLP-1 normalization. A commercial probiotic mixture called VSL#3, which contained four strains of *Lactobacillus*, three strains of *Bifidobacterium*, and one strain of *Streptococcus*, boosted butyrate levels, which in turn promoted the production of GLP-1 in a mouse model. GLP-1 level normalization was associated with decreased meal consumption and enhanced glucose tolerance [131]. Also, Pegah et al. demonstrated that probiotics, as well as resveratrol, decreased glucose and IR in diabetic rats by increasing GLP-1 levels and reducing oxidative stress [132]. Han and colleagues reported that the process by which fecal microbiota transplantation from a normal glucose tolerance donor improved glycolipid abnormality in db/db mice involved modifications to the bacterial composition that generates SCFAs and triggers the GPR43/GLP-1 pathway [133]. For an exhaustive review on the mechanisms, applications, and challenges of probiotics useful for counteracting T2D, see Shen et al. [134].

### 4.2. Antioxidants

IR and hyperglycemia—which is brought on by oxidative stress, disrupting glucose homeostasis—can both contribute to T2D [135]. Antioxidants, both dietary and endogenous, can alter gut microbiota composition and function, improving metabolic health. Dietary polyphenols, which are naturally occurring substances found in fruits, vegetables, cereals, tea, coffee, and wine, play an important role in the interaction between exogenous antioxidants and the gut microbiota, regulating the incidence and progression of T2D. Because of this, the small intestine absorbs only a tiny portion (5–10%) of the total amount of polyphenols consumed. Approximately 90–95% of the total amount of polyphenols consumed are deposited in the large intestine, where they are broken down into smaller phenolic metabolites by the gut microbial population. Consuming meals high in polyphenols may have health advantages since these metabolites are absorbable [136]. Research has shown that higher consumption of polyphenol-rich foods is linked to improved glycemic control and reduced inflammation in patients with type 2 diabetes (T2D) [135]. Furthermore, the intake of polyphenols and the primary foods that contain them may help to lower IR and reduce risk factors for T2D, including oxidative stress and inflammation, as evidenced by numerous human studies.

The pertinent research relating dietary polyphenols to prediabetes and T2D is reviewed by Guasch-Ferrè et al. [137], who concentrated on a number of clinical trials examining the impact of polyphenols on cardiometabolic parameters in T2D patients. For instance, in a meta-analysis including data from 22 randomized clinical trials, catechins from green tea were found to have a positive effect on decreasing fasting glucose; however, its effects on fasting insulin, HbA1c, and the Homeostatic Model Assessment for IR (HOMA-IR) were not significant.

Green tea has been shown to correct microbial dysbiosis associated with various conditions, including obesity, which negatively impacts the onset of T2D. In particular, the abundance of the phylum *Bacteroidetes* significantly decreased in obese mice, whereas that of the phylum *Firmicutes* significantly increased. Observational studies have indicated that regular consumption of green tea is associated with improved glycemic control and a lowered risk of T2D [138]. Furthermore, the administration of green tea was able to influence the composition of the gut microbial community by elevating the relative abundances of some advantageous (and adversely correlated with obesity) bacteria, such *Akkermansia*, *Lachnospiraceae*, and *Alistipes* [139].

Numerous extracts rich in polyphenol content have shown antidiabetic effects. Rutin is the primary flavonoid glycoside found in *djulis husk* crude extract, which was the source of one such polyphenol. In order to evaluate the preventive impact of rutin and *djulis husk* crude extract on glucose tolerance, mice were given a high-fat diet (HFD) for 16 weeks in order to cause hyperglycemia. Tung et al. found that the crude extract significantly reduced HFD-induced diabetogenic effects. Furthermore, the crude extract markedly elevated the phosphorylation of insulin receptor substrate 1 (pIRS1) and glucose transporter type 4 (GLUT4) protein expression in epididymal white adipose tissue and the liver. Moreover, the crude extract restored the HFD-induced decrease in catalase (CAT) and glutathione peroxidase (GPX) activities [140]. Furthermore, in HFD-fed animals, the expression of Zonula occludens-1 (ZO-1) and occludin proteins in the colon was significantly downregulated, reducing intestinal permeability and LPS translocation. However, these effects were reverted after the application of the crude extract. Furthermore, the crude extract intervention significantly influenced the gut microbiota community and its alpha diversity. Therefore, in cases of hyperglycemia brought on by a high-fat diet, the crude extract of *djulis* shell may raise blood glucose levels and improve insulin receptor sensitivity. Its capacity to control the gut microbiota, preserve intestinal barrier integrity to lower body inflammation, boost antioxidant activity, and alter insulin signaling may be the reason for this action [140].

*Glycyrrhiza uralensis* polysaccharide extract (GUP) is another extract that shows promise for improving the gut microbiota in connection with T2D. It has exceptional antioxidant properties, inhibits alpha-glucosidase activity, and enhances glycemic management in T2D. The genus *Glycyrrhiza* has many health benefits and contains bioactive substances that are important for T2D treatment. A study showed that *G. uralensis* polysaccharide extract (GUP) reduced liver lipid levels, oxidative stress, IR, and high blood sugar in T2 diabetic mice. GUP also improved gut microbiota diversity, reducing harmful species such as *Bacteroides*, *Escherichia-Shigella*, and *Clostridium sensu stricto 1*, and increasing beneficial ones like *Akkermansia*, *Lactobacillus*, *Romboutsia*, and *Faecalibaculum*. This research provides valuable information on dietary approaches for managing T2D and promoting overall well-being [141].

Melatonin interacts with gut bacteria to reduce T2D as well. Huang et al.’s case–control research examined the relationship between T2D risk and serum melatonin in a community of southern Chinese people, emphasizing the importance of the gut flora. Higher blood melatonin levels were linked to a decreased risk of T2D and lower levels of fasting glucose, according to research including 2034 people with T2D (cases) and healthy people (controls). A reduced risk of T2D was associated with greater blood melatonin levels. The gut microbiota and melatonin signaling mediated the connection; tryptophan metabolites may play a particular role in this process. These results highlight the significance of melatonin and associated microorganisms and metabolites as possible targets for T2D treatment. The study also discovered that the gut microbiota of people with T2D was altered, showing decreased levels of serum melatonin, a less diverse gut microbiota, a greater abundance of *Bifidobacterium*, and a lower abundance of *Coprococcus*. Seven genera were also discovered to be linked. Furthermore, a correlation between melatonin and features associated with T2D was discovered to exist for seven genera. Among these, serum LPS and interleukin-10 (IL-10) showed positive correlations with *Bifidobacterium*, whereas serum interleukin-1β (IL-1β), interleukin-6 (IL-6), interleukin-10 (IL-10), interleukin-17 (IL-17), α tumor necrosis factor TNF-α, and LPS showed negative correlations with *Coprococcus* [142]. It is worth mentioning a study that investigated the impact of a dietary intervention involving high-fiber, polyphenol-rich, and vegetable protein functional foods on the fecal microbiota and various biochemical parameters in patients with T2D. These parameters included LPS, BCAAs, TMAO, HbA1c, and FFAs.

Patients with T2D showed imbalances in their intestinal bacteria, with an increase in the abundance of *Prevotella copri*. Dietary intervention with functional foods significantly enhanced the fecal microbiota compared with the placebo, leading to increased alpha diversity and modifying the abundances of specific bacteria. This change occurred regardless of any T2D medication they were taking. The patients had fewer *P. copri* bacteria and more of the bacteria *F. prausnitzii* and *A. muciniphila*, which are known to have anti-inflammatory effects. The group who changed their diet also saw significant reductions in glucose, total and low-density lipoprotein (LDL) cholesterol, FFAs, HbA1c, triglycerides, and CRP, as well as an improvement in antioxidant activity, compared to the group who did not change their diet. Long-term adherence to a high-fiber, polyphenol-enriched, and vegetable protein-based diet provided benefits for gut microbiota composition and may offer potential therapies for to improve glycemic control, dyslipidemia, and inflammation [143].

### 4.3. Chinese Medicine

Traditional Chinese medicine (TCM) has emerged as a promising approach to the management of T2D [144,145]. Because TCM is mostly taken orally, it has a direct impact on the gut microbiome. Nevertheless, a number of active components found in herbal remedies, including tannins, flavonoids, and triterpene glycosides, have characteristics such as high molecular flexibility, low lipophilicity, and hydrogen bonding capability, which restrict TCM bioavailability. Its pharmacology is significantly influenced by absorption, which is facilitated by a change in the gut microbiota. Furthermore, because different TCM components function as nutrients for the growth of particular bacteria, TCM modulates the population of host intestinal microbes. T2D can be treated with a particular Chinese medicine formula called Huang-Qi-Ling-Hua-San (HQLHS). *Astragalus Membranaceus*, *Ganoderma lucidum*, *Inonotus obliquus*, and *Momordica charantia* make up its composition. Using a T2D mouse model induced by streptozotocin and a high-fat diet, researchers were able to better understand how HQLHS reduced high blood sugar and high cholesterol levels. HQLHS enhanced the abundances of *Bifidobacterium*, *Turicibacter*, *Alistipes*, *Romboutsia*, and *Christensenella*. Next, fecal microbiota transplantation (FMT) was employed to determine whether these bacteria had any therapeutic value, with a focus on two particular strains of *Christensenella*. The results demonstrated that feeding these strains to diabetic rats improved their ability to metabolize fat and blood sugar, raised GLP-1 production, strengthened their antioxidant defenses, and induced other positive benefits. Overall, the evidence pointed to *Christensenella* being a major target for HQLHS treatment with a beneficial effect on T2D [146].

The fruit of *Lycium barbarum* is another plant used in traditional Chinese medicine. It is abundant in flavonoids, which have strong exploration potential and are linked to anti-inflammatory and antioxidant properties that may lower the incidence of T2D. In T2D mice induced with a high-fat diet (HFD) and streptozotocin (STZ), its therapeutic impact was examined. Significant antidiabetic activity was demonstrated by the study, which included decreased water intake, liver index, fasting blood glucose, HOMA-IR, HOMA-IS, HbA1c levels, and Oral Glucose Tolerance Test (OGTT) levels, along with improved lipid and glucose metabolism. Additionally, the study demonstrated the restoration of liver tissue structure, as evidenced by a decrease in fat vesicles, and a remission of hepatocyte swelling, as well as a reduction in the expression of proinflammatory cytokines and related mRNAs. The study also looked at alterations in the gut microbiome of mice with T2D induced by HFD/STZ. Bacteria including *Bacteroidales*_S24-7_group, *Allobaculum*, *Turicibacter*, *Coriobacteriaceae*, *Ruminococcaceae*, *Clostridiales*_vadinBB60_group, and *Enterococcus* were shown to be common in T2D patients. These findings showed improvements in the metabolic profile, gut microbiota health, and glucose levels [147].

In a separate study, Wang et al. administered oral treatments for constipation to HFD-fed mice and db/db mice using the Chinese formula Shouhuitongbian (SHTB; 200 and 100 mg/kg/d) and the standard medication metformin (100 mg/kg/d). The results showed that SHTB successfully reduced inflammation, improved dysfunctional lipid metabolism, and enhanced IR and glucose tolerance. Furthermore, SHTB demonstrated effectiveness in treating T2D by altering the composition of the gut microbiota, particularly by increasing the abundances of *Akkermansia* and *Parabacteroides*, and by promoting the production of SCFAs and the breakdown of BCAAs [148].

**Table 1 nutrients-16-03951-t001:** Supplements demonstrating efficacy T2D management.

Supplements	Model	Administration (Dose/Day, Duration) of Supplemetation	General Effects and Possible Mechanism of Action	Effect on Microbiome Composition	Refs.
Probiotics					
	HeLa cell line	1–2 h treatment	↓ translocation of NF-kB to the nucleus and expression of inflammatory cytokines	↑ *Lactobacillus* and blocked metabolic endotoxemia via NF-kB activation	[128]
Fermented milk with *Lactobacillus acidophilus* La-5 and *Bifidobacterium animalis* subsp *lactis* BB-12	T2D patients	120 g/d for 6 weeks	Improved glycemic profile: ↓ fructosamine and HbA1c levels		[129]
*Akkermansia muciniphila*	Male C57BL/6 *ob/ob*, HF-fed obese, and type 2 diabetic mice	4 weeks	Improved metabolic profile: ↓ fat-mass gain, metabolic endotoxemia, adipose tissue inflammation, and IR. ↑ levels of endocannabinoids in the intestine.		[130,131]
*VSL#3*	*Animal model*: C57J/B6 male mice (n = 7 in each group) 4–6-week-old male mice fed with a low-fat diet or a high-fat diet, with and without VSL#3. In vitro *model:* NCI-H716 cells line	5 mg/kg body weight by oral gavage for 8 weeks	Using a cell culture system, it has been demontrated that butyrate stimulated the release of GLP-1 from intestinal L-cells. Potential therapeutic utility to counter obesity and T2D.	↑ levels of a short-chain fatty acid (SCFA), butyrate.	[131]
Antioxidants—Polyphenols					
Green Tea	Obese mice	Regular consumption of green tea	Improved glycemic control	↓ *Bacteroidetes* and ↑ *Firmicutes*; ↑ *Akkermansia*, *Lachnospiraceae*, and *Alistipes*	[138,139]
*Djulis hull* crude extract—rutin	HFD mice with hyperglycemia	16 weeks	↑ pIRS1 and GLUT4 protein expression in eWAT and liver; ↓ AUC, OGTT, HOMA-IR, and AGE levels; ↓ CAT and GPX activities; upregulated expression of ZO-1 and occluding proteins in the colon	↑ microbial community and alpha diversity; preserved intestinal barrier integrity	[140]
*Glycyrrhiza uralensis* polysaccharide extract	HFD/STZ-induced T2D mice	400 mg/kg for 4 weeks	↓ liver lipid levels, oxidative stress, IR, and hyperglycemia	↑ alpha diversity; enhanced intestinal barrier integrity; ↓ *Bacteroidetes*, *Escherichia*-*Shigella*, and *Clostridium*, and ↑ *Akkermansia*, *Lactobacillus*, *Romboutsia*, and *Faecalibaculum*	[141]
Dietary intervention with functional food	81 T2D patients	3 months	↓ glucose, total and LDL cholesterol, free fatty acids, HbA1c, triglycerides, and CRP; ↑ antioxidant activity, glycemic control, dyslipidemia, and inflammation	↑ alpha diversity; ↓ *Prevotella copri* (increased in T2D patients), and ↑ *Faecalibacterium prausnitzii* and *Akkermansia muciniphila*	[143]
Chinese medicine					
Huang-Qi-Ling-Hua-San (HQLHS)	T2D mouse	300 ul once a day	Improved fat and blood sugar metabolism, ↑ GLP-1 production, strengthened their antioxidant defenses	↑ *Bifidobacterium*, *Turicibacter*, *Alistipes*, *Romboutsia*, and *Christensenella*	[146]
Flavonoids from *Lycium barbarum* (LBFs)	HFD/STZ-induced T2D mice		Antidiabetic activity: ↓ water intake, liver index, fasting blood glucose, HOMA-IR, HOMA-IS, HbA1c levels, and OGTT levels; ↓ proinflammatory cytokine expression		[147]
Shouhuitongbian (SHTB)	HFD-fed mice and db/db mice	200 and 100 mg/kg/d	↓ inflammation, repaired dysfunctional lipid metabolism, and improved IR and glucose intolerance; IRS-1/PI3K/AKT signaling pathway overexpression	↑ *Akkermansia* and *Parabacteroides*; ↓ BCAAs and ↑ BSCFAs and SCFAs	[148]

Abbreviations. AGE: advanced glycation end product; AUC: area under the curve; BCAAs: branched-chain amino acids; BSCFAs: branched short-chain fatty acids; CAT: catalase; CRP: C-reactive protein; eWAT: epididymal white adipose tissue; GLP-1: glucagon-like peptide 1; GLUT4: glucose transporter type 4; GPX: glutathione peroxidase; HbA1c: glycosylated hemoglobin A1C; HFD/STZ: streptozotocin high-fat diet; HFD: high-fat diet; HOMA-IR: Homeostatic Model Assessment for Insulin Resistance; HOMA-IS: Homeostatic Model Assessment for Insulin Sensivity; HQLHS: Huang-Qi-Ling-Hua-San; IR: insulin resistance; LBFs: flavonoids from *Lycium barbarum*; LDL: low-density lipoprotein; NCI-H716: cell line derived from ascites fluid of a colorectal adenocarcinoma from a 33-year-old Caucasian male; NF-kB: nuclear factor kappa-light-chain-enhancer of activated B cells; *ob/ob* mice: obese mice; OGTT: Oral Glucose Tolerance Test; PI3K: phosphatidylinositol 3-kinase; pIRS1: phosphorylation of insulin receptor substrate 1; SCFA: short-chain fatty acid; SHTB: Shouhuitongbian; T2D: type 2 diabetes; VSL#3: Probiotic “Very Safe Lactobacilli#3”; ZO-1: zonula occludens-1; ↓: Decrease; ↑: Increase.

## 5. Physical Exercise, T2D, and the Gut Microbiota

Physical activity may have a protective effect on gut health and T2D. Not only is exercise often recommended for weight loss and maintenance, but some studies suggest that both acute and chronic exercise may reduce the risk of fatty liver and gastrointestinal disorders, as well as inflammatory mediators and apoptotic markers in gut lymphocytes, as observed in older animals [149]. This review looked at research exploring how exercise might protect against or reduce mechanisms associated with the microbiota that are believed to be responsible for triggering the processes underlying T2D.

### 5.1. The Role of Physical Activity in Intestinal Barrier Function and Metabolic Health

Yu et al. [150] recently investigated the relationship between intestinal permeability and physical activity. Specifically, they showed that aerobic exercise, through its ability to induce a redistribution of hypoxia-inducible factor-1 α (HIF-1α), modulated the transcription of many barrier-protective genes and anti-inflammatory and tissue-protective pathways [151]. In this context, chronic aerobic exercise stimulated AMP-activated protein kinase (AMPK), an important downstream signaling pathway of sextrin2 (SESN2)-mediated activity [152,153,154,155,156,157]. The latter is a stress-inducible protein that promotes epithelial cell survival and recovery and acts as a positive regulator of exercise-induced improvements in glycolipid metabolism [151]. SESN2 is critical in controlling different types of stressors, allowing epithelial cells to recover from inflammatory damage [153,154,155]. According to research by Yu et al. [150], chronic aerobic exercise significantly reduced weight gain induced by a high-fat diet, improved body composition in mice, and increased fasting glucose levels. Notably, the levels of pro-inflammatory cytokines (TNF-α, IL-1β, and IL-6) in the control and SENS2 ablation groups were significantly reduced with chronic aerobic exercise [158]. Interestingly, chronic aerobic exercise restored the levels of alpha diversity by increasing the abundance of *Bacteroidetes* and reducing the abundances of *Firmicutes* and *Actinobacteria* that the high-fat diet (HFD) had significantly increased in the mice [150]. Moreover, regular exercise tends to increase several genera of bacteria, contributing to improved gut health and metabolic benefits. *Bifidobacterium*, *Lactobacillus*, and *Akkermansia* are associated with enhanced metabolic health. The abundance of *Bacteroides*, which play a role in breaking down complex molecules in the gut, also increases with exercise. Additionally, *Roseburia*, known for producing short-chain fatty acids (SCFAs) that support gut health, tends to proliferate with physical activity [159,160]. In this way, exercise significantly protected the intestinal epithelium from bacterial invasion. In addition, the colon villus musculature thickened significantly after chronic aerobic exercise, showing a large increase in TJ protein expression, as well as improved motility, as evidenced by more frequent stools. Although these are early findings, the results on the role of chronic aerobic exercise in protecting the gut barrier are promising and exciting.

### 5.2. The Importance of Chronic Endurance Exercises in SCFA Production and Consumption for the Prevention of T2D

Numerous studies have examined the different functions of SCFAs in IR and T2D. These functions include controlling immunomodulatory processes, maintaining intestinal epithelial integrity, and controlling insulin secretion and pancreatic cell proliferation [161]. Owing to the therapeutic functions of SCFAs, studies on how exercise affects the production and consumption of these fats were conducted [162,163]. Specifically, Allen J.M. [164] studied whether a 6-week aerobic exercise program may alter gut microbial populations and fecal SCFAs in previously inactive, lean, and obese adults. The authors found that fecal concentrations of SCFAs (acetate, propionate, and butyrate) increased mostly in the lean subjects after three sessions per week (30 to 60 min) of moderate-to-vigorous aerobic exercise over a period of six weeks, and this effect appeared to be dependent on BMI status. After six weeks of returning to sedentary habits, the concentrations of SCFAs in the obese group remained unchanged. Nevertheless, the washout period was associated with a decline in the gut microbiome’s capacity to produce SCFAs in the obese group, as evidenced by a decrease in the relative abundances of butyrate- and propionate-regulating genes [164]. These results, although interesting, need further study with exercise protocols of greater duration, intensity, and complexity. Finally, with the current state of the art, we can conclude that, although the effect on obese people was not as strong as that on thin people, it appears that constant physical activity plays a significant role in the regulation of SCFAs and that the latter has a direct relationship with T2D.

The abundances of several genera of bacteria that produce SCFAs tend to increase after exercise. These include *Roseburia*, known for producing butyrate, a type of SCFA that supports gut health; *Faecalibacterium*, another butyrate producer, which is often associated with anti-inflammatory effects; *Akkermansia*, primarily known for its role in maintaining the gut lining, but also contributes to SCFA production; and *Bifidobacterium*, which produces acetate, another important SCFA [159,160].

### 5.3. Exercise Modulation in the Reduction of LPS

As previously described (Section 2.1), LPS causes metabolic endotoxemia and IR and represents a trigger for chronic low-grade inflammation through the LPS–CD14 pathway. Following the consumption of high-fat, high-carbohydrate meals, LPS plasma concentrations greatly increase [165], indicating that the source of this fat-soluble LPS is the gastrointestinal tract. Furthermore, dietary fat has been demonstrated to raise intestinal permeability to LPS [166]. In a study [167], researchers examined the hypothesis that a regimen of swimming for 60 minutes per session, 5 days per week for 8 weeks, ameliorates insulin resistance in high-fat diet (HFD)-induced obese (DIO) rats through the regulation of TAK1-dependent signaling and its hepatic regulators. HFD feeding led to increased body weight, visceral fat mass, serum free fatty acids (FFAs), and hepatic lipid deposition, while reducing hepatic glycogen content and insulin sensitivity. Both chronic and acute exercise training improved insulin resistance. Exercise training resulted in decreased phosphorylation of TAK1, c-Jun N-terminal kinase 1 (JNK1), and insulin receptor substrate 1 (IRS1), while enhancing Akt phosphorylation in the liver. Furthermore, exercise elevated the protein levels of USP4 and DUSP14 and reduced the protein levels of TRIM8 in the liver of obese rats. These findings demonstrate that exercise induces significant modulation of TAK1-dependent signaling and its regulators in the liver, leading to marked improvements in insulin sensitivity. This study provides novel insights into the mechanisms by which physical exercise mitigates insulin resistance. Exercise robustly reversed the activation of this pathway and enhanced insulin signaling, presenting a novel mechanism by which exercise improves insulin action in obesity and T2D. Thus, exercise, both acute and chronic, promoted a reduction in serum LPS in rats that had been induced to eat HFD. This pathway has been validated in various models examining the impact of exercise on LPS-induced lung inflammation [168] and LPS-induced inflammatory responses in rat cardiac tissue [169]. In conclusion, the available in vivo and in vitro evidence supports the need for clinical studies to investigate the effects of moderate- and low-intensity exercise on different inflammatory LPS-induced conditions.

### 5.4. The Effect of Exercise in Counteracting the Production of TMAO Associated with T2D

Recent results suggest that food has little bearing on the positive effects of regular exercise on gut microbial populations [170,171,172]. Consequently, physical activity may encourage the production of less dangerous bioactive metabolites, such as TMAO, by improving the profile of the gut microbiome, especially in older or obese people who are more prone to have a dysfunctional gut microbiota [173]. Argyridou et al. [174] analyzed baseline and 12-month follow-up data from the Walking Away from T2D study, which recruited adults at high risk for T2D in primary care in 2009–2010. During this period, 316 men and 167 women were analyzed. Moderate to vigorous physical activity (about 30 min per day) was associated with TMAO in all models. The results showed that each 30 min difference in moderate to vigorous physical activity was associated with less TMAO, while sedentariness and light physical activity were not associated with TMAO in any model. Thus, engagement in physical activity was associated with lower TMAO levels, suggesting a possible new mechanism underlying the inverse relationship between physical activity and cardiometabolic health [174]. In addition, Battillo and Malin [175] demonstrated that a low-calorie diet plus high-intensity interval exercise intervention reduced TMAO more than a low-calorie diet-only program. The study involved 23 sedentary women with obesity (age: 48.4 ± 2.4 years; BMI: 37.9 ± 1.4 kg/m^2^), who were randomized to complete 12 supervised high-intensity interval exercise sessions (3 min 50% peak heart rate alternating with 3 min at 90% for 60 min) over 13 days. The exercise duration was progressively increased so that participants completed 30 and 45 min of interval training on the first and second days, respectively, and subsequently 60 min of exercise per session, with 1 rest day over the 13 days. Overall, only in women with higher levels of circulating TMAO at baseline did both treatments reduce plasma TMAO [176]. Finally, according to the above findings, Erickson et al. demonstrated that the change in TMAO after diet and exercise intervention was inversely related to visceral adipose tissue at baseline (r = −0.63, *p* = 0.009) and glucose disposal rates (r = 0.58, *p* = 0.002) [176]. In conclusion, we can state that a low-calorie diet and exercise approach appears to be effective in reducing TMAO.

### 5.5. The Effect of Exercise on GLP-1 and the Gut Microbiota in T2D

Exercise influences GLP-1 through several mechanisms. Physical activity stimulates the secretion of GLP-1 from intestinal L cells, partly due to the increased production of SCFAs by beneficial gut bacteria, which are more abundant in physically active individuals [159,160,177]. Exercise also enhances insulin sensitivity, amplifying the effects of GLP-1, which helps to regulate blood glucose levels by promoting insulin secretion and inhibiting glucagon release [178]. Additionally, regular physical activity increases the population of bacteria that produce SCFAs, which not only stimulates GLP-1 secretion but also improves gut barrier function and reduce inflammation [177]. Moreover, exercise aids in weight loss and maintenance, enhancing the effectiveness of GLP-1 by reducing IR and inflammation [179]. These mechanisms collectively contribute to better glycemic control and overall metabolic health in individuals with T2D. Different types of exercise influence GLP-1 levels in various ways, enhancing its beneficial effects on glucose metabolism and overall health. Aerobic exercises, such as running and cycling, significantly increase GLP-1 secretion by improving gut microbiota composition and increasing the production of SCFAs [180]. Resistance training, including weight-lifting and body weight exercises, boosts GLP-1 levels by enhancing muscle insulin sensitivity and promoting lean muscle mass, which is crucial for glucose uptake [180]. High-intensity interval training (HIIT) is particularly effective in rapidly increasing GLP-1 levels due to its intense bursts of activity followed by short rest periods, which improve cardiovascular fitness and insulin sensitivity [181]. These exercises collectively contribute to better glycemic control, weight management, and overall metabolic health in individuals with T2D [182].

## 6. Conclusions

Although the important role of hereditary components in determining the onset of T2D is acknowledged, the disease primarily results from the common sedentary lifestyle and the sharp rise in obesity worldwide. T2D comorbidities can cause early death, hasten the course of the disease, and induce mental and physical illnesses. These comorbidities have an adverse effect on general health, well-being, exacerbating diabetes-related outcomes, and related expenditures in patients with T2D, leading to a substantial treatment burden, higher healthcare utilization, costs, and lost economic output.

T2D is influenced by the gut microbiota, which also affects IR, glucose homeostasis, and inflammation, major contributors to the development of T2D. It also affects how the intestinal tract and extraintestinal tissues respond to antidiabetic medications. In this review, we focused on the role that lifestyle plays in maintaining a healthy microbiota and the implications for the development and treatment of T2D. Nonetheless, further clinical research is required to elucidate the possible therapeutic impact of gut microbes and their byproducts on T2D. Given the complexity of the variables that may affect clinical outcomes, including basal conditions, nutrition, lifestyle, and medications, translating the understanding of the microbiota for clinical benefit in diabetics is both a challenging and exciting task for scientists. Figure 1 summarizes the factors that predispose or aggravate T2D and those that improve the situation, also acting through the microbiota, highlighting the central role of GLP-1. Emphasizing the need for a tailored approach is crucial, particularly when addressing a diverse and specific group of individuals, which includes pre-diabetics, confirmed patients, and those at risk of T2D. In order to minimize side effects and increase benefits, the same level of attention to individual features—made possible by technical progress—should be given to both identifying pharmaceutical interventions and non-pharmacological ones.

A diet that is tailored to individual needs, whether it includes supplements or not, a well thought out and meticulously implemented exercise chronic regimen, and customized drug therapies, paying particular attention to gut microbiota status and changes, represents an integrated strategy against T2D. In addition, it is important to keep an eye on how things are developing to guarantee that the appropriate course of action is taken. This review, far from being exhaustive, aims to provide useful insights for the implementation of an integrated and individualized strategy for T2D prevention and management.

## Figures and Tables

**Figure 1 nutrients-16-03951-f001:**
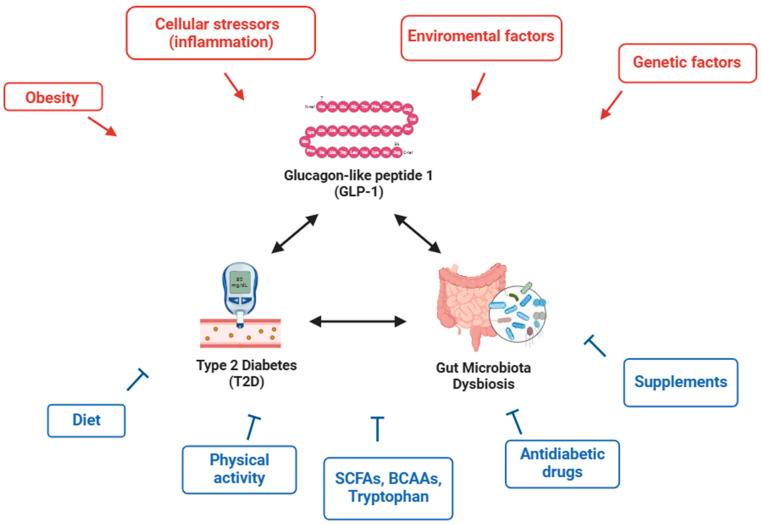
An integrated approach to maintaining a healthy gut microbiota for T2D prevention and management.

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
