# Peer review of "Targeting the Gut Microbiota for Prevention and Management of Type 2 Diabetes"

_nutrients, 2024, doi:10.3390/nu16223951_

Round 1
Reviewer 1 Report
Comments and Suggestions for Authors
This paper comprehensively reviews the relationship between gut microbiota and T2D, emphasizing how factors like diet, supplements, and physical exercise impact metabolic health. It effectively underscores the therapeutic potential of modulating gut microbiota and metabolites as strategies for T2D management, supported by recent studies and illustrative examples. This approach provides valuable insights into developing integrative, microbiota-centered treatments for T2D. Below are specific comments for further consideration:
1_2.1. Gut Barrier Permeability and Inflammation
Expanding on the research background for Akkermansia muciniphila as a "next-generation probiotic" would clarify its significance in this section.
2_2.2. Short-Chain Fatty Acids (SCFAs)
Adding details on specific butyrate-producing bacteria, such as Faecalibacterium prausnitzii and Roseburia intestinalis, could illustrate how these bacteria contribute to insulin sensitivity, gut barrier integrity, and overall metabolic health, enhancing understanding of SCFAs' role in T2D prevention and treatment.
3_2.3. Branched-Chain Amino Acids (BCAAs)
Including specific examples of microbiota would improve clarity, especially regarding BCAAs and Bacteroides. For instance, the study by N. Yoshida et al. (iScience 24(11), 2021, 103342) illustrates how Bacteroides spp. promotes BCAA catabolism in brown fat and reduce obesity risk. Such references could help readers appreciate the role of specific bacterial species in BCAA metabolism and their metabolic health impacts.
4_2.4. Bile Acids (BAs)
The authors need to consider that secondary bile acids (BAs) may be more cytotoxic than primary BAs. Secondary BAs, such as deoxycholic acid (DCA) and lithocholic acid (LCA), produced by gut bacteria, may damage cell membranes, induce DNA damage, and trigger inflammation, especially at high concentrations. This toxicity is also linked to an increased cancer risk. Including this information would provide a balanced view of secondary BAs' dual roles in gut health.
5_2.7. Antidiabetic Drugs and Gut Microbiota
Removing "gut microbiota" from the title could make it more concise and better aligned with prior sections. This would shift focus to antidiabetic drugs' relationship with metabolic products in T2D management and create a smoother content flow.
6_3. Diet and T2D
Dividing this section into subsections could clarify each dietary strategy's effects and significance, making it easier for readers to compare them. Such an organization would highlight each diet's unique features and benefits, enhancing overall comprehension.
7_5. Physical Exercise and T2D
This section lacks specific microbiota names. To clarify the impact of exercise on gut microbiota, adding specific bacteria that increase or decrease with physical activity would make the content more detailed and easier to understand.
Author Response
Manuscript ID: nutrients-3306030
|
Response to Reviewer 1 Comments
|
||
|
1. Summary |
|
|
|
Thank you very much for your careful revision and valuable comments, which have greatly contributed to improving the manuscript. We have addressed all requests, highlighting any revisions, and have provided a document for each reviewer detailing, point by point, the changes made to the manuscript. We believe that the paper is now more complete, interesting, and easier to read. Please find the detailed responses below, with the corresponding revisions and corrections highlighted in the re-submitted files.
|
||
|
2. Point-by-point response to Comments and Suggestions for Authors
|
||
|
Comments 1: Expanding on the research background for Akkermansia muciniphila as a "next-generation probiotic" would clarify its significance in this section. |
||
|
Response 1: Thank you for pointing this out. The research background for Akkermansia muciniphila as a "next-generation probiotic" has been better explained, highlighting its role in promoting gut epithelial health and in reducing inflammation, clarifying its significance in this section
|
||
|
Comments 2: Adding details on specific butyrate-producing bacteria, such as Faecalibacterium prausnitzii and Roseburia intestinalis, could illustrate how these bacteria contribute to insulin sensitivity, gut barrier integrity, and overall metabolic health, enhancing understanding of SCFAs' role in T2D prevention and treatment. |
||
|
Response 2: Agree. We have, accordingly, revised to emphasize this point. This section has been modified adding further details on SCFAs’ role in T2D prevention and treatment and on specific butyrate-producing bacteria, as suggested
Comments 3: Including specific examples of microbiota would improve clarity, especially regarding BCAAs and Bacteroides. For instance, the study by N. Yoshida et al. (iScience 24(11), 2021, 103342) illustrates how Bacteroides spp. promotes BCAA catabolism in brown fat and reduce obesity risk. Such references could help readers appreciate the role of specific bacterial species in BCAA metabolism and their metabolic health impacts. Response 3: Thanks for this helpful suggestion, the importance of BCAA in T2D has been emphasized and the above mentioned paper has been summarized to better explain how Bacteroides can influence BCAA metabolism, underscoring the importance of specific bacterial species in metabolic health.
Comments 4: The authors need to consider that secondary bile acids (BAs) may be more cytotoxic than primary BAs. Secondary BAs, such as deoxycholic acid (DCA) and lithocholic acid (LCA), produced by gut bacteria, may damage cell membranes, induce DNA damage, and trigger inflammation, especially at high concentrations. This toxicity is also linked to an increased cancer risk. Including this information would provide a balanced view of secondary BAs' dual roles in gut health. Response 4: As suggested, the cytotoxic effect of some secondary bile acids has been described, evidencing the secondary BAs' dual roles in gut health.
Comments 5: Removing "gut microbiota" from the title could make it more concise and better aligned with prior sections. This would shift focus to antidiabetic drugs' relationship with metabolic products in T2D management and create a smoother content flow. Response 5: Thanks for this comment, “Gut microbiota” from the title has been removed
Comments 6: Dividing this section into subsections could clarify each dietary strategy's effects and significance, making it easier for readers to compare them. Such an organization would highlight each diet's unique features and benefits, enhancing overall comprehension. Response 6: Thanks for this comment, this section has been divided into subsections as suggested by the referee
Comments 7: This section lacks specific microbiota names. To clarify the impact of exercise on gut microbiota, adding specific bacteria that increase or decrease with physical activity would make the content more detailed and easier to understand. Response 7: We appreciate the reviewer's insightful comment. In response, we have revised the section to include specific microbiota names. We have added details on the bacteria that increase or decrease with physical activity, providing a clearer and more detailed explanation of the impact of exercise on gut microbiota.
|
||

Reviewer 2 Report
Comments and Suggestions for Authors
The authors of this narrative review explore the role of gut dysbiosis in Type 2 Diabetes (T2D), with a specific focus on how lifestyle factors, such as diet and exercise, impact T2D prevention and management. This is a highly relevant topic; however, it’s worth noting that numerous recent manuscripts have also investigated this area. Below are my suggestions to strengthen the manuscript:
1.Abstract: It would be beneficial for the authors to highlight that cardiovascular complications are a primary concern in T2D management.
2.Line 56: Specific types of DM, such as those caused by genetic factors, exocrine pancreatic disorders, and medication-induced cases (incorporate these data).
3.Lines 64-67: It would improve clarity to list the most significant complications of T2D as microvascular and macrovascular diseases, followed by cancer and other issues. Please specify if these complications may arise before T2D develops. The same adjustment is suggested for lines 69-70: hereditary factors should precede other considerations.
4.Chronic Low-Grade Inflammation: The authors should ensure consistent terminology by referring to “chronic low-grade inflammation” uniformly throughout the text (e.g., lines 91, 167, 707).
5.GLP-1 Receptor Agonists (GLP-1 RAs): Please specify which GLP-1 RAs are referenced, such as liraglutide and others.
6.Introduction - Aim: Adding a brief statement on the aim of this review at the end of the introduction would help clarify the focus of the manuscript for readers.
7.Section 2.1: Consider defining “dysbiosis” and clarify that you refer to the “leaky gut” hypothesis.
8.Line 712: Please confirm whether you are referring to TLR4 or TLR5.
9.General Suggestions:
-Use “DM” as the abbreviation for diabetes mellitus throughout the manuscript (e.g., replace “diabetes” on lines 52, 55, 60, etc.).
- Add a list of abbreviations specific to Table 1, as well as a general list of abbreviations for the entire manuscript.
- Review and potentially revise section titles for improved clarity, ensuring they adequately reflect content related to both microbiota and T2D.
- Consider renaming Section 6 to “Conclusions” for consistency.
- Address minor typographical errors throughout, such as on line 132.
- Enhance the academic tone and language throughout the manuscript. In addition, consider revising paragraph structures to improve readability, as the current use of frequent, short paragraphs could be streamlined.
Comments on the Quality of English LanguageEnhance the academic tone and language throughout the manuscript.
Author Response
Manuscript ID: nutrients-3306030
|
Response to Reviewer 2 Comments
|
||
|
1. Summary |
|
|
|
Thank you very much for your careful revision and valuable comments, which have greatly contributed to improving the manuscript. We have addressed all requests, highlighting any revisions, and have provided a document for each reviewer detailing, point by point, the changes made to the manuscript. We believe that the paper is now more complete, interesting, and easier to read. Please find the detailed responses below, with the corresponding revisions and corrections highlighted in the re-submitted files.
|
||
|
3. Point-by-point response to Comments and Suggestions for Authors
|
||
|
Comments 1: Abstract: It would be beneficial for the authors to highlight that cardiovascular complications are a primary concern in T2D management. |
||
|
Response 1: Thanks for the suggestion, the fact that cardiovascular complications are a primary concern in T2D management and prognosis has been highlighted in the Abstract.
|
||
|
Comments 2: Line 56: Specific types of DM, such as those caused by genetic factors, exocrine pancreatic disorders, and medication-induced cases (incorporate these data). |
||
|
Response 2: Agree. Different diabetes types in this section have been described.
Comments 3: Lines 64-67: It would improve clarity to list the most significant complications of T2D as microvascular and macrovascular diseases, followed by cancer and other issues. Please specify if these complications may arise before T2D develops. The same adjustment is suggested for lines 69-70: hereditary factors should precede other considerations. Response 3: Thanks for this helpful suggestion, the sentences have been revised in order to list complications and factors as suggested and the period in which comorbidities can arise has been better specified.
Comments 4: Chronic Low-Grade Inflammation: The authors should ensure consistent terminology by referring to “chronic low-grade inflammation” uniformly throughout the text (e.g., lines 91, 167, 707). Response 4: Thanks for this comment. As suggested, The paper has been corrected in order to ensure consistent terminology by referring to “chronic low-grade inflammation” uniformly throughout the text
Comments 5: GLP-1 Receptor Agonists (GLP-1 RAs): Please specify which GLP-1 RAs are referenced, such as liraglutide and others. Response 5: Thanks for this comment, which type of GLP-1 RAs are referenced has been specified, as requested.
Comments 6: Introduction - Aim: Adding a brief statement on the aim of this review at the end of the introduction would help clarify the focus of the manuscript for readers. Response 6: Thanks for this comment, A statement has been added at the end of introduction to better clarify the aim of the review, as suggested.
Comments 7: Section 2.1: Consider defining “dysbiosis” and clarify that you refer to the “leaky gut” hypothesis. Response 7: We appreciate the reviewer's insightful comment. Dysbiosis” has been defined and the fact that we refer to the “leaky gut” hypothesis, defining it, has been specified, enhancing overall comprehension.
|
||
Comments 8: Line 712: Please confirm whether you are referring to TLR4 or TLR5.
Response 8: Thanks for this comment. We confirm that it is TLR4.
Comment 9: General Suggestions:
- Use “DM” as the abbreviation for diabetes mellitus throughout the manuscript (e.g., replace “diabetes” on lines 52, 55, 60, etc.).
Response 1: Thanks for this comment. We replaced diabetes with DM where appropriate.
- Add a list of abbreviations specific to Table 1, as well as a general list of abbreviations for the entire manuscript.
Response 2: Thanks for this suggestion. We have added a list of abbreviations specific to Table 1, and a general list of abbreviations for the entire manuscript.
- Review and potentially revise section titles for improved clarity, ensuring they adequately reflect content related to both microbiota and T2D.
Response 3: We have reviewed and revised section titles as suggested.
- Consider renaming Section 6 to “Conclusions” for consistency.
Response 4: Thanks for this comment. We have changed the chapter title.
- Address minor typographical errors throughout, such as on line 132.
Response 5: We have addressed minor typographical errors throughout, as suggested.
- Enhance the academic tone and language throughout the manuscript. In addition, consider revising paragraph structures to improve readability, as the current use of frequent, short paragraphs could be streamlined.
Response 6: We enhanced the academic tone and language throughout the manuscript, with particular attention to abstract and conclusion, and we revised paragraph structures.
